# Movement Penalized Bayesian Optimization with Application to Wind Energy Systems

**Shyam Sundhar Ramesh**
ETH Zurich
shramesh@ethz.ch

**Pier Giuseppe Sessa**
ETH Zurich
sessap@ethz.ch

**Andreas Krause**
ETH Zurich
krausea@ethz.ch

**Ilija Bogunovic**
University College London
i.bogunovic@ucl.ac.uk

## Abstract

Contextual Bayesian optimization (CBO) is a powerful framework for sequential decision-making given side information, with important applications, e.g., in wind energy systems. In this setting, the learner receives context (e.g., weather conditions) at each round, and has to choose an action (e.g., turbine parameters). Standard algorithms assume no cost for switching their decisions at every round. However, in many practical applications, there is a cost associated with such changes, which should be minimized. We introduce the episodic CBO with movement costs problem and, based on the online learning approach for metrical task systems of Coester and Lee [19], propose a novel randomized mirror descent algorithm that makes use of Gaussian Process confidence bounds. We compare its performance with the offline optimal sequence for each episode and provide rigorous regret guarantees. We further demonstrate our approach on the important real-world application of altitude optimization for Airborne Wind Energy Systems. In the presence of substantial movement costs, our algorithm consistently outperforms standard CBO algorithms.

## 1 Introduction

Bayesian optimization (BO) is a well-established framework for sequential black-box function optimization that relies on Gaussian Process (GP) models [42] to sequentially learn and optimize the unknown objective. In many practical scenarios, however, one wants to additionally use available *contextual* information when making decisions. In this setting, at each round, the learner receives a context from the environment and has to choose an action based upon it. Previous works have developed contextual BO algorithms [32, 16, 31, 40], and applied them to various important applications, e.g., vaccine design, nuclear fusion, database tuning, crop recommender systems, etc.

A potential practical issue with these standard algorithms is that they assume no explicit costs for *switching* between their actions at every round. Frequent action changes can be extremely costly in many real-world applications. This work is motivated by the problem of real-time control of the altitude of an airborne wind energy (AWE) system.[1] In AWE systems, the wind speed is often only measurable at the system's altitude, and determining the optimal operating altitude of an AWE system as the wind speed varies represents a challenging problem. Another fundamental challenge is that additional energy is required for adjusting the altitude, which makes the frequent altitude changes costly. Consequently, this work is motivated by the following question: *How can we efficiently learn*

---

[1]AWE system is a wind turbine with a rotor supported in the air without a tower that can benefit from the persistence of wind at different high altitudes [22].

*to optimize the AWE system's operating altitude despite varying wind conditions while minimizing the energy cost associated with turbine altitude changes?*

In this work, we formalize the *movement penalized* contextual BO problem. When the switching cost is a *metric* (distance function), we propose a novel algorithm that effectively combines ideas from BO with the online learning strategies proposed in [19] for solving the so-called *metrical task system* (MTS) problem [11]. Furthermore, our algorithm relies solely on noisy point evaluations (i.e., bandit feedback), allows for arbitrary context sequences, and besides the standard exploration-exploitation trade-off, it also balances the movement costs. As a result, it outperforms the standard movement-cost-agnostic contextual BO algorithms as well as movement-conservative baselines (see Fig. 1).

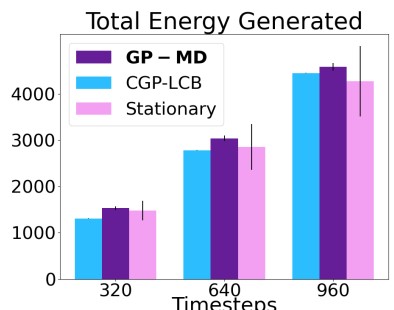

Figure 1: Total energy generated by the AWE system when operated with GP-MD (proposed in this work), CGP-LCB [32] and a stationary baseline. The stationary baseline employs no-learning and does not incur movement costs. GP-MD and CGP-LCB learn an operating strategy, but GP-MD outperforms CGP-LCB since it also considers movement costs.

**Related Work.** Bayesian optimization (BO) refers to a sequential approach for optimizing an unknown objective (cost function) from noisy point evaluations. A great number of BO methods have been developed over the years (e.g., [38, 50, 17]). While the focus of standard BO approaches is mainly on optimizing the unknown objective cost function, in this work, we additionally focus on penalizing frequent action changes. This problem makes the most sense in the *contextual* BO setting [32], where the main objective changes with the observed contextual information and the learner also seeks to minimize the cost associated with frequent changes of its actions. Several works have tried to incorporate such and similar cost functions in the BO setting. [35] explicitly consider switching costs based on how deep into the pipeline the change in variable occurs. But they do not consider the contextual setting that is essential for our wind energy application and hence our work is not directly comparable with theirs. Moreover, it assumes that the system of interest has a modular structure (as detailed in [35, Section-3]). In such a modular setting, the cost at each round is the number of modules that has to be changed rather than the amount of change of each variable. Other works include cost-aware and multi-objective sampling strategies in various settings such as batch [29, 27], multi-fidelity [10, 28, 49], multi-objective optimization [2] and dynamic programming [34, 33]. Finally, two recent works [24, 15] consider the problem of switching cost minimization in Bayesian optimization. They both consider the non-contextual setting, and while [24] lacks theoretical guarantees, [15] does not explicitly consider the cost in the regret definition. Similarly, we consider movement/switching costs, but unlike these previous work, we specifically focus on minimizing the movement costs in the contextual setting.

The *Metrical Task Systems* (MTS) problem [11] is a sequential decision-making problem widely studied in the *online learning* literature. At each round, the learner observes a service cost function, chooses an action, and incurs the corresponding cost together with a movement cost penalizing the distance (according to some metric) between the current action and the one chosen at the previous round. The MTS problem is directly related to our problem setting (see Section 2.2). After a long series of works, in [14] and [19], an $\mathcal{O}\big((\log n)^2\big)$-competitive algorithm for MTS on any finite metric space was established. The approach of Coester and Lee [19] relies on a tree representation of the decision space and an action randomization scheme via a mirror descent procedure. In contrast to our setting, these works assume that the service costs are *known* (i.e., they assume the *full-information* feedback). Moreover, in the online optimization literature, other related works study the online *convex* optimization with switching costs [26, 48, 25, 36] and convex body chasing problems [12, 5, 43, 13]. We make no use of convexity and take a model-based (GP) approach to learn about the unknown service costs. [37] consider non-convex objective functions, however, they impose certain conditions on the objective that are not easily modeled via GP as done in our work.

The use of GP confidence bounds in online learning settings has been previously explored, e.g., in repeated multi-agent [44, 47] and sequential games [46], and to discover randomized max-min strategies [45]. However, none of these works has considered movement costs in the objective. This makes our problem significantly different from the aforementioned ones, and requires a suitable action randomization scheme that can trade-off exploration, exploitation, and movement costs simultaneously.

Various works have applied Bayesian Optimization in the context of *wind energy systems* before. [41, 21, 4] use BO techniques to maximize the total energy yield in a wind farm in a cooperative or a closed-loop framework. [39] use it to tune the parameters of the wind turbine to learn the effective wind speed. [6] use BO for plant design (i.e., physical system design) of airborne wind energy systems. [51] apply BO and other regression techniques to predict short-term wind energy production to make informed production offers. Finally, [3] survey several different methods for efficient wind-power prediction using machine-learning methods including BO. Our problem formulation differs from the above mentioned works, since we explicitly consider movement energy loss caused by the altitude change of the system.

**Contributions.** We formally introduce Bayesian optimization with movement costs and propose a novel GP-MD algorithm (in Algorithm 1). GP-MD combines the online mirror descent (MD) algorithm with shrinking Gaussian Process (GP) confidence bound to decide on which point to evaluate next. In our theoretical analysis, we establish rigorous sublinear regret guarantees for our algorithm by combining techniques from Bayesian optimization and metrical task systems approaches [19]. Finally, we demonstrate that GP-MD is able to successfully outperform previous contextual Bayesian optimization approaches on both synthetic and real-world data in the presence of movement costs. In particular, we consider the application to airborne wind energy systems and demonstrate that GP-MD can effectively operate such a system by considering movement costs and varying environmental conditions.

## 2 Problem Statement

Let $f : \mathcal{X} \times \mathcal{E} \to \mathbb{R}_+$ be an *unknown* cost function defined over $\mathcal{X} \times \mathcal{E} \subset \mathbb{R}^p$, where $\mathcal{X}$ is a finite set of actions, i.e., $|\mathcal{X}| = n$, and $\mathcal{E}$ represents convex and compact space of contexts. We denote the *known* metric (i.e., distance function) of $\mathcal{X}$ as $d(\cdot, \cdot)$, and similarly to other works in Bayesian optimization (e.g., [50, 17]) assume that the target cost function $f$ belongs to a reproducing kernel Hilbert space (RKHS) $\mathcal{H}_k$ of functions (defined on $\mathcal{X} \times \mathcal{E}$), that corresponds to a known kernel $k : (\mathcal{X} \times \mathcal{E}) \times (\mathcal{X} \times \mathcal{E}) \to \mathbb{R}_+$ with $k((x, e), (x', e')) \leq 1$ for any action-context pair. In particular, we assume that for some known $B > 0$, the target cost $f$ has a bounded RKHS norm, i.e., $f \in \mathcal{F}_k = \{f \in \mathcal{H}_k : \|f\|_k \leq B\}$. Also, we assume that the diameter of $\mathcal{X}$ ($\max_{x,x' \in \mathcal{X}} d(x, x')$) is bounded and denote it by $\psi$.

We consider an episodic setting, wherein each episode runs over a finite time horizon $H$. Let the *initial state* of the system in the first episode correspond to action $x_{0,1} \in \mathcal{X}$. At the end of every episode, the system resets to a new given initial action $x_{0,m} \in \mathcal{X}$ where $m \in \{1, 2, \ldots, N_{ep}\}$ denotes the episode index. In each episode $m$ and at every time step $h \in \{1, 2, \ldots, H\}$, the environment reveals the context $e_{h,m} \in \mathcal{E}$ to the learner. We make no assumptions on the context sequence provided by the environment (i.e., it can be arbitrary and different across episodes). The learner then chooses $x_{h,m} \in \mathcal{X}$ and observes the noisy function value:

$$y_{h,m} = f(x_{h,m}, e_{h,m}) + \xi_{h,m}, \tag{1}$$

where $\xi_{h,m} \sim \mathcal{N}(0, \sigma^2)$ with known $\sigma$, and independence over time steps. The goal of the learner is to minimize the cost incurred over the rounds in every episode, but at the same time to minimize the distance between its subsequent decisions as measured by $d(x_{h-1,m}, x_{h,m})$.

Let $D_m = \{x_{1,m}, x_{2,m}, \ldots, x_{H,m}\}$ denote the set of actions chosen by the learner over $H$ rounds in episode $m$. We recall that each action $x_{h,m} \in D_m$ is chosen after observing the corresponding context $e_{h,m}$. The objective is to minimize the cumulative episodic cost for each episode $m$,

$$\text{cost}_m(D_m) = \underbrace{\sum_{h=1}^{H} f(x_{h,m}, e_{h,m})}_{S_m(D_m)} + \underbrace{\sum_{h=1}^{H} d(x_{h,m}, x_{h-1,m})}_{M_m(D_m)}, \tag{2}$$

where we refer to the two terms in Eq. (2) as *service cost* $S_m$ and *movement cost* $M_m$.

When $f$ is known, the problem can be seen as a MTS instance as detailed in Section 2.2. Even in such a case, we cannot hope to solve this problem optimally, and nearly-optimal approximate algorithms were recently proposed (see Coester and Lee [19]). Hence, the learner's performance in episode $m$ is measured via $(\alpha, \beta)$-approximate regret:

$$r_m^{\alpha,\beta} = \text{cost}_m(D_m) - \alpha \cdot \text{cost}_m(D_m^*) - \beta, \tag{3}$$

where $D_m^* := \arg\min_{D \subset \mathcal{X}, \; |D|=H} \text{cost}_m(D)$ is the offline optimal action sequence obtained assuming the knowledge of the true sequence of contexts $\{e_{h,m}\}_{h=1}^H$ *in advance*, and $\alpha$ and $\beta$ are approximation constants (independent of $N_{ep}$). In contrast, in our setting, the learner only gets to see the *current context* when making a decision and has no knowledge about the future ones.

After $N_{ep}$ episodes, the total cumulative regret is defined as

$$R_{N_{ep}}^{\alpha,\beta} = \sum_{m=1}^{N_{ep}} r_m^{\alpha,\beta}. \tag{4}$$

We seek an algorithm whose total cumulative regret grows sublinearly in $N_{ep}$, so that $\lim_{N_{ep} \to \infty} R_{H,N_{ep}}^{\alpha,\beta}/N_{ep} = 0$, for any set of initial states $\{x_{0,m}\}_{m=1}^{N_{ep}} \subset \mathcal{X}$.

## 2.1 Gaussian Process Model

In standard Bayesian optimization, a surrogate Gaussian Process model is typically used to model the target cost function. A Gaussian Process $GP(\mu(\cdot), k(\cdot, \cdot))$ over the input domain $\mathcal{X} \times \mathcal{E}$, is a collection of random variables $(f(x,e))_{x \in \mathcal{X}, e \in \mathcal{E}}$ where every finite number of them $(f(x_i, e_i))_{i=1}^n$, $n \in \mathbb{N}$, is jointly Gaussian with mean $\mathbb{E}[f(x_i)] = \mu(x_i, e_i)$ and covariance $\mathbb{E}[(f(x_i, e_i) - \mu(x_i, e_i))(f(x_j, e_j) - \mu(x_j, e_j))] = k((x_i, e_i), (x_j, e_j))$ for every $1 \leq i, j \leq n$.

BO algorithms typically use zero-mean GP priors to model uncertainty in $f$, i.e., $f \sim GP(0, k(\cdot, \cdot))$, and Gaussian likelihood models for the observed data. As more data points are observed, GP (Bayesian) posterior updates are performed in which noise variables are assumed to be drawn independently across $t$ from $\mathcal{N}(0, \lambda)$. Here, $\lambda$ is a hyperparameter that might be different from the true noise variance $\sigma^2$. More precisely, given a sequence of previously queried points and their noisy observations the posterior is again Gaussian, with the posterior mean and variance given by:

$$\mu_t(x,e) = k_t(x,e)^T (K_t + \lambda I_t)^{-1} Y_t \tag{5}$$

$$\sigma_t^2(x,e) = k((x,e),(x,e)) - k_t(x,e)^T (K_t + \lambda I_t)^{-1} k_t(x,e), \tag{6}$$

where $Y_t := [y_1, \ldots, y_t]$ denotes a vector of observations, $K_t = [k((x_s, e_s), (x_{s'}, e_{s'}))]_{s,s' \leq t}$ is the corresponding kernel matrix, and $k_t(x,e) = [k((x_1, e_1), (x,e)), \ldots, k((x_t, e_t), (x,e))]^T \in \mathbb{R}^{t \times 1}$.

**Maximum Information Gain.** In the standard Bayesian optimization, the main quantity that characterizes the complexity of optimizing the target cost function is the maximum information gain [50] defined at time $t$ as:

$$\gamma_t = \max_{\{(x_i, e_i)\}_{i=1}^t} I(Y_t; f), \tag{7}$$

where $I(Y_t; f)$ denotes the mutual information between random observations $Y_t$ and GP model $f$. The mutual information for the GP model is given as:

$$I(Y_t; f) = \frac{1}{2} \log \det(I_t + \lambda^{-1} K_t). \tag{8}$$

This quantity is kernel-specific and for compact and convex domains $\gamma_t$ is sublinear in $t$ for various classes of kernel functions [50] as well as for kernel compositions (e.g., additive kernels in [32]).

**Confidence Bounds.** We also use the following result ([50, 1, 17]) that is frequently used in Bayesian optimization to provide confidence bounds around the unknown function.

**Lemma 1.** *Assume the $\sigma$-sub-Gaussian noise model as in Eq. (1), and let $f$ belong to $\mathcal{F}_k$. Then, the following holds with probability at least $1 - \delta$ simultaneously over all $t \geq 1$ and $x \in \mathcal{X}$, $e \in \mathcal{E}$:*

$$|\mu_t(x,e) - f(x,e)| \leq \beta_t \sigma_t(x,e), \tag{9}$$

*where $\beta_t = \frac{\sigma}{\lambda^{1/2}} \sqrt{2 \ln(1/\delta) + 2\gamma_t} + B$, and $\mu_t$ and $\sigma_t$ are defined in Eqs. (5) and (6) with $\lambda > 0$.*

Based on the previous, we also define the lower confidence bound for every $x \in \mathcal{X}, e \in \mathcal{E}$ as:

$$\text{lcb}_t(x,e) := \mu_t(x,e) - \beta_t \sigma_t(x,e). \tag{10}$$

We use $\text{lcb}_m(x,e)$ when it is computed based on data collected before episode $m$.

---
**Algorithm 1** GP-MD
---
1: **Require:** Action space $\mathcal{X}$, kernel function $k(\cdot, \cdot)$, metric $d(\cdot, \cdot)$
2: Run FRT($\mathcal{X}, d(\cdot, \cdot)$) and obtain $\tau$-HST $\mathcal{T} = (V, E)$ with leaves $\mathcal{L} = \mathcal{X}$
3: **for** $m = 1, \ldots, N_{ep}$ **do**
4:      Receive $x_{0,m}$ and initialize $z_{0,m}$ (Eq. (29)), conditional prob. $q_0 = \Delta^{-1}(z_{0,m})$ as in Eq. (15)
5:      **for** $h = 1, \ldots, H$ **do**
6:          Observe context $e_{h,m}$ and initialize costs: $\mathrm{lcb}_m(v, e_{h,m}) = 0, \forall v \in V \backslash \mathcal{L}$
7:          **for** $u \in \mathcal{OD}(V \backslash \mathcal{L})$ **do**
8:              Update vertex prob. $q_h^{(u)}$ from $q_{h-1}^{(u)}$ and $\mathrm{lcb}_m(\cdot, e_{h,m})$ via Mirror Descent (Eq. (13))
9:              Update cost for vertex $u$:

$$\mathrm{lcb}_m(u, e_{h,m}) = \langle q_h^{(u)}, \mathrm{lcb}_m(\cdot, e_{h,m}) \rangle = \sum_{\nu \in \mathcal{C}(u)} q_{h,\nu} \cdot \mathrm{lcb}_m(\nu, e_{h,m})$$

10:          **end for**
11:          Compute prob. vector $z_{h,m} = \Delta(q_h)$ (Eq. (15)) and leaves' prob. $l(z_{h,m})$ (Eq. (11))
12:          Estimate optimal coupling $\zeta_{h-1,h,m}$ between $l(z_{h-1,m})$ and $l(z_{h,m})$ as in Eq. (12)
13:          Sample action $x_{h,m} \sim \zeta_{h-1,h,m}(\cdot | x_{h-1,m})$ and observe $y_{h,m} = f(x_{h,m}, e_{h,m}) + \xi_{h,m}$
14:      **end for**
15:      Update $\mu_{m+1}(\cdot, \cdot)$ and $\sigma_{m+1}(\cdot, \cdot)$ as per Eq. (5) and Eq. (6)
16: **end for**
---

## 2.2 Relation to Metrical Task Systems (MTS)

When $f$ is known, our optimization objective in Eq. (2) can be seen as a particular type of MTS problem, where $f(\cdot, e_{h,m})$ is the MTS service cost that changes for every $h$ and $m$. Compared to a standard MTS (see Appendix A), our problem formulation is more challenging since the learner can only learn about $f$ from previously observed data. The approach proposed in this paper builds on the algorithm by Coester and Lee [19] for standard MTS problems. However, to cope with the aforementioned challenge, our approach exploits the regularity assumptions regarding $f$ and utilizes the constructed lower confidence bounds Eq. (10) to *hallucinate* information about the unavailable service cost at each round. Before presenting our overall approach, we describe a preliminary step proposed by [19], which consists of representing our metric space $(\mathcal{X}, d)$ by a *Hierarchically Separated Tree* (HST) metric space.

**HST metric space.** Consider a tree $\mathcal{T} = (V, E)$ with root $r$, leaves $\mathcal{L} \subset V$ and non-negative weights $w_v$, for each $v \in V$, which are non-increasing along root-leaf paths. Let $d_{\mathcal{T}}(l, l')$ denote a distance metric between any two leaves $l, l' \in \mathcal{L}$ given as the sum of the encountered weights on the path from $l$ to $l'$ (see Fig. 4). $(\mathcal{L}, d_{\mathcal{T}})$ is a HST metric space, and $\tau$-HST metric space if the weights are exponentially decreasing, i.e., $w_u \leq w_v / \tau$, with $v$ being the parent of $u$.

Similarly to [19], we use the algorithm from [23] (which we name via the authors' surnames as FRT) to approximate the given metric space $(\mathcal{X}, d)$ by a $\tau$-HST one. In particular, we use FRT in Algorithm 1 as a computationally efficient preprocessing step to create a tree $\mathcal{T}$ with leaves $\mathcal{L}$ corresponding to actions in $\mathcal{X}$, distance metric $d_{\mathcal{T}}$, and root node $r$. We explain the intrinsic MTS motivation for this preprocessing step in Appendix B.1, and defer additional details to Appendix B.2.

## 3 The GP-MD Algorithm

In this section, we introduce GP-MD, a novel algorithm for the contextual BO problem with movement costs defined in Section 2. At each episode $m$ and round $h$, the state of GP-MD can be summarized by a vector of probabilities $z_{h,m} \in K_{\mathcal{T}}$ over the vertices of $\mathcal{T}$, where $K_{\mathcal{T}} := \Big\{ z \in \mathbb{R}_+^{|V|} : z_r = 1, z_u = \sum_{\nu \in \mathcal{C}(u)} z_\nu \quad \forall u \in V \backslash \mathcal{L} \Big\}$, and $\mathcal{C}(u)$ denotes the children of $u$. Each entry $z_\nu$ represents the probability that the selected action $x_{h,m}$ belongs to the leaves of the subtree rooted at $\nu$, i.e., $z_\nu = \mathbb{P}(x_{h,m} \in \mathcal{L}(\nu))$. Below, we specify how $z_{h,m}$ is computed at each round. Moreover,

given any $z \in K_\mathcal{T}$, the vector

$$l(z) := [z_l, \ l \in \mathcal{L}] \in [0, 1]^n, \tag{11}$$

defines a probability distribution over the leaves $\mathcal{L}$, and hence the actions $\mathcal{X}$. As in [19], given probability vectors $z_{h,m}$ and $z_{h-1,m}$, GP-MD computes the *minimal distance* distribution

$$\zeta_{h-1,h,m} = \underset{\zeta \in \Pi(l(z_{h-1,m}), l(z_{h,m}))}{\arg \inf} \mathbb{E}_\zeta[d_\mathcal{T}(U_{h-1,m}, U_{h,m})], \tag{12}$$

where $U_{h-1,m}$ and $U_h$ are random variables having marginals $l(z_{h-1,m})$ and $l(z_{h,m})$ respectively. Finally, action $x_{h,m}$ is sampled from the conditional minimal distance distribution $x_{h,m} \sim \zeta_{h-1,h,m}(\cdot|x_{h-1,m})$ (Line 13 in Algorithm 1). At the end of each episode $m$, the newly observed data are then used to update posterior mean and standard deviation about the cost function.

Finally, we describe how probability vectors $z_{h,m}$ are computed at each round, a key step of GP-MD (Lines 8–12 in Algorithm 1). We follow the recursive Mirror Descent (MD) procedure proposed by [19], with the important difference that we are dealing with an *unknown* context-dependent cost function. Hence, we make use of the Gaussian process model and corresponding confidence estimates as defined in Section 2.1.

To obtain probabilities $z_{h,m}$, we consider *conditional* probability vectors $q \in Q_\mathcal{T}$, where $Q_\mathcal{T}$ is the set of valid conditional probabilities $Q_\mathcal{T} := \left\{ q \in \mathcal{R}_+^{|V \setminus r|} : \sum_{\nu \in \mathcal{C}(u)} q_\nu = 1 \quad \forall u \in V \setminus \mathcal{L} \right\}$. For each vertex $\nu$ with parent $u$, $q_\nu$ represents the conditional probability $\mathbb{P}(x_{h,m} \in \mathcal{L}(\nu)|x_{h,m} \in \mathcal{L}(u))$. Moreover, given $q_h \in Q_\mathcal{T}$ we define the vector $q_h^{(u)} := [q_{h,\nu}, \ \nu \in \mathcal{C}(u)]$ as the conditional distribution over children of $u$, and let $Q_\mathcal{T}^{(u)}$ be the set of all valid distributions $q_h^{(u)}$.

In each episode $m$, conditional probability vector $q_h$ for round $h$ is obtained recursively, from leaves to root, as a function of $q_{h-1}$, the observed context $e_{h,m}$, and the current estimate about the cost associated to each particular vertex. More precisely, let $\mathcal{OD}(V \setminus \mathcal{L})$ be a topological ordering of the internal vertices $V \setminus \mathcal{L}$ so that every child in $\mathcal{T}$ occurs before its parent. Then, for each $u \in \mathcal{OD}(V \setminus \mathcal{L})$ conditional probabilities $q_h^{(u)}$ are obtained via the Mirror Descent update:

$$q_h^{(u)} = \underset{p \in Q_\mathcal{T}^{(u)}}{\arg \min} \left\{ D^{(u)}(p \| q_{h-1}^{(u)}) + \langle p, \mathrm{lcb}_m^{(u)}(\cdot, e_{h,m}) \rangle \right\}. \tag{13}$$

Function $D^{(u)}$ is the Bregman divergence with respect to a suitable potential function (see Appendix D.1), while $\mathrm{lcb}_m^{(u)}(\cdot, e_{h,m}) := [\mathrm{lcb}_m(\nu, e_{h,m}), \ \forall \nu \in \mathcal{C}(u)]$ is a lower confidence bound estimate of the costs corresponding to children of vertex $u$. For $v \in \mathcal{L}$, $\mathrm{lcb}_m(\nu, e_{h,m})$ are obtained by the GP-regression techniques outlined in Section 2.1, while for internal vertices these are computed recursively from their children nodes as:

$$\mathrm{lcb}_m(u, e_{h,m}) := \sum_{\nu \in \mathcal{C}(u)} q_{h,\nu} \mathrm{lcb}_m(\nu, e_{h,m}). \tag{14}$$

The movement cost is primarily controlled by this usage of Bregman divergence based mirror descent. Also, sampling from the conditional minimal distance distribution further restricts movement between alternate actions. Once the vector of conditional probabilities $q_h \in Q_\mathcal{T}$ has been updated, we can obtain the corresponding probability vector $z_{h,m}$ via the mapping $\Delta : Q_\mathcal{T} \to K_\mathcal{T}$ such that:

$$z = \Delta(q) \implies z_\nu = z_u q_\nu \quad \forall u \in V \setminus \mathcal{L}, \ \nu \in \mathcal{C}(u). \tag{15}$$

### 3.1 Theoretical Guarantees

Our main theorem bounds the cumulative regret of GP-MD.

**Theorem 1.** *Let $\mathcal{X}$ be represented by a $\tau$-HST space with $\tau > 4$ (Line 2 of Algorithm 1), and set $\delta \in (0, 1)$. Then, with probability at least $1 - \delta$, the regret of GP-MD over $N_{ep}$ episodes is bounded by*

$$R_{N_{ep}}^{\alpha, \beta} = \mathcal{O}\left( \beta_{N_{ep}} \left( N_{ep} H^2 \gamma_{H N_{ep}} + H \log(\tfrac{H}{\delta}) \right)^{\frac{1}{2}} + H(B + \psi) \log \left( \tfrac{N_{ep} \log(N_{ep})}{\delta} \right) \right),$$

*with approximation factors $\alpha = \mathcal{O}\big((\log n)^2\big)$ and $\beta = \mathcal{O}(1)$. Here, $H$ is the episodes' length, $\beta_{N_{ep}}$ is the confidence level from Lemma 1, and $\gamma_{H N_{ep}}$ is the maximum information gain defined in Eq. (7).*

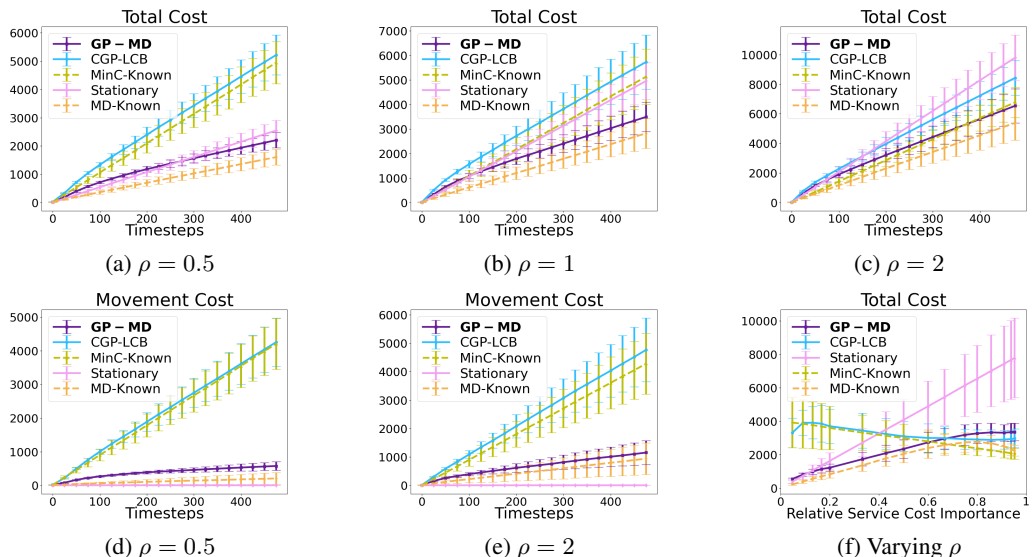

Figure 2: Total and movement cost performance of algorithms on synthetic functions for varying importance of movement/service cost (i.e., different $\rho$ values). GP-MD outperforms CGP-LCB in terms of total incurred cost, and its performance closely follows one of the idealized benchmark MD-KNOWN. GP-MD also minimizes the movement cost while CGP-LCB suffers from significant movements (Figs. 2d and 2e). The performance of GP-MD remains robust when the movement cost importance in the total cost objective diminishes (Fig. 2f).

For most of the popularly used kernels, Thm. 1 can be made more explicit by substituting bounds on $\gamma_{HN_{ep}}$ (e.g., in the case of linear kernel and compact domain, we have $\gamma_t = O(p \log t)$, while for squared-exponential kernel it holds $\gamma_t = O((\log t)^{p+1})$ [50]; see also Section 2.1). In such cases, we make the following two important observations regarding our result: i) The obtained regret bound is sublinear in the number of episodes $N_{ep}$ and hence $\lim_{N_{ep} \to \infty} R_{N_{ep}}^{\alpha, \beta}/N_{ep} = 0$; ii) The bound is independent of the input space size, i.e., the number of actions $n$ (although the approximation factor $\alpha$ depends logarithmically on $n$, similarly to [19]). These imply that GP-MD approaches $\alpha$-competitive ratio performance of the MTS algorithm by [19], while learning about the service cost from noisy point evaluations (i.e., bandit feedback) only. Finally, in our analysis, we treat $H$ as constant.

Proof of Thm. 1 is detailed in Appendix E. Next, we outline some main steps. We make use of the competitive ratio guarantees for the used Mirror Descent algorithm from [19, Corollary 4] to bound the expected hallucinated service cost $\sum_{h=1}^{H} \langle \text{lcb}_m(\cdot, e_{h,m}), l(z_{h,m}) \rangle$. Here, we use the fact that $\text{lcb}_m(\cdot, \cdot)$ does not change within an episode. Since this is not the actual service cost, we bound $\sum_{h=1}^{H} \langle f(\cdot, e_{h,m}), l(z_{h,m}) \rangle$ by $\sum_{h=1}^{H} \langle \text{lcb}_m(\cdot, e_{h,m}), l(z_{h,m}) \rangle$ with an additional learning error. The sum of the learning errors over all episodes can be rewritten as $\sum_{m=1}^{N_{ep}} \sum_{h=1}^{H} \langle \sigma_m^2(\cdot, e_{h,m}), l(z_{h,m}) \rangle$. We use the concentration of the conditional mean result from [30, Lemma 3] to upper bound it by the actual realizations $\sum_{m=1}^{N_{ep}} \sum_{h=1}^{H} \sigma_m^2(x_{h,m}, e_{h,m})$, and use the result of [18, Lemma-2], to further upper bound it with the maximum information gain quantity $\gamma_{N_{ep}H}$.

Finally, the movement cost can also be bounded similar to that of the previously mentioned expected hallucinated service cost $\sum_{h=1}^{H} \langle \text{lcb}_m(\cdot, e_{h,m}), l(z_{h,m}) \rangle$ with an extra $\alpha = \mathcal{O}((\log n)^2)$ factor using [19, Corollary 4].

## 4 Experiments

This section provides numerical results on synthetic and real-world data. We compare the performance of our GP-MD algorithm with the following baselines:

- STATIONARY selects the stationary strategy $x_h = x_0$ for all $h$,
- CGP-LCB [32] neglects the movement cost and sets $x_h = \arg\min_x \text{lcb}_h(x, e_h)$ for all $h$,
- MINC-KNOWN assumes $f(\cdot)$ is known and chooses $x_h = \arg\min_x f(x, e_h)$, and
- MD-KNOWN assumes $f(\cdot)$ is known and runs mirror descent from [19] on $f(\cdot, e_h)$.

MD-KNOWN and MINC-KNOWN unrealistically assume that $f(\cdot)$ is known and can be seen as upper-bound for the achievable performance of GP-MD and CGP-LCB, respectively. We use the same constant value $\beta = 2.0$ for the exploration parameter in both GP-MD and CGP-LCB (since the theoretical worst-case bounds are found to be overly pessimistic and can impede performance [50]). We run the algorithms over a single episode (as done in CGP-LCB). We also discover that updating the confidence bounds after every timestep in GP-MD leads to better performance in practice.

**Synthetic experiments.** We consider synthetic experiments, where the objective function is a random GP sample. The considered action space $\mathcal{X}$ is a subset of $[0, 1]^2$ consisting of 400 points that form the uniform grid, while the context space $\mathcal{E}$ consists of 40 contexts that are uniformly sampled from $(0, 1)$. We sample objective function (i.e., actual cost) $f : \mathcal{X} \times \mathcal{E} \rightarrow \mathbb{R}$ from a $GP(0, k)$, where $k$ is a squared exponential kernel with lengthscale parameter set to $l = 0.2$. We use the Euclidean distance between the domain points as the movement cost, calculate the distance matrix (between every pair of points) and the average movement cost. We subtract the minimum value from $f$ and scale it such that the average function value is equal to the average movement cost. We also set the noise parameter to $1\%$ of the function range. We introduce the trade-off parameter $\rho \in \{0.25, 0.5, 1, 2, 4\}$ that only multiplies the service cost, i.e, $\rho f(x, e)$, but not the movement cost. This is to test the performance of the algorithms for varying importance of the service/movement costs. For each $\rho$ we sample 25 different functions and run the algorithms for 500 timesteps wherein at each step the contexts are randomly sampled.

In Fig. 2a-Fig. 2e, we show the total cumulative cost as a function of timesteps for different $\rho$ values. Then, in Fig. 2f, we show the performance of the algorithms (for known kernel parameters) when run for 800 timesteps for varying importance of the service and movement costs. In particular, we consider a convex combination of the service and movement costs, where we set the respective coefficients multiplying these two objectives as $\rho/(1 + \rho)$ and $1/(1 + \rho)$.

As shown in Fig. 2a-Fig. 2c, the performance of GP-MD is generally close to the one of the idealized, unrealistic benchmark MD-KNOWN, which, as expected, performs the best. The stationary baseline performs comparably when $\rho$ is small, while its performance deteriorates for larger values. As expected, both MINC-KNOWN and CGP-LCB incur higher total costs than GP-MD when the movement cost is of the higher or same relative importance as the service cost (i.e., $\rho \in \{0.5, 1.0\}$), while the performance gap slowly decreases when the service cost becomes dominant ($\rho = 2.0$). In Figs. 2d and 2e, we also show the corresponding movement costs, and observe that movement cost ignorant CGP-LCB incurs significant movement costs, while our GP-MD successfully minimizes the movement costs. Finally, in Fig. 2f, we observe that the performance of GP-MD is robust, i.e., it clearly outperforms CGP-LCB whenever the movement cost dominates the total cost objective, while its performance remains comparable to the one of CGP-LCB (that is built to minimize service cost) when the movement cost becomes dominated by the service cost.

### 4.1 Altitude Optimization in AWE Systems

In airborne wind energy (AWE) systems, the turbine's operating altitude can be changed depending on the wind pattern. We follow the setup of Baheri et al. [7] that applied CGP-LCB [32] which ignores movement-costs, to learn this control task. In this section, we use a dataset from [9] which contains wind-speed information over various locations in Europe for a period ranging from 2011 to 2017, and also includes measurements at different altitudes per location. We consider the wind speed data from the second half of 2016. Our goal is to maximize the generated energy, while taking into account the energy loss due to moving the turbine from one altitude to another.

We consider 25 different altitudes (ranging from 10m to 1600m) as the action space and the context space to be hours in the day (i.e., 24 values). We define our unknown service objective function to be $f(x, t) = \max_{x'}(E_S(x', t)) - E_S(x, t)$ where $E_S(x, t)$ denotes the energy generated based on the windspeed at altitude $x$ and time $t$. Based on the discrete-time power generation formula from Baheri et al. [7] (Eq. (10)), we have

$$E_S(x, t) = \left(c_1(\min\{V_w(x, t), V_r\})^3 - c_2 V_w^2(x, t)\right)\Delta t. \tag{16}$$

Here $V_w(x, t)$ denotes the windspeed at altitude $x$ and time $t$, and $V_r$ denotes the rated windspeed of the turbine. The constants $c_1 = 0.0579$ and $c_2 = 0.09$ are system dependent. This corresponds to the energy generated at a particular altitude $x$ for $\Delta t$ time. Similarly to Baheri et al. [7], we use $\Delta t = 60$ since we consider intervals of one hour length. Next, we define the movement cost to be

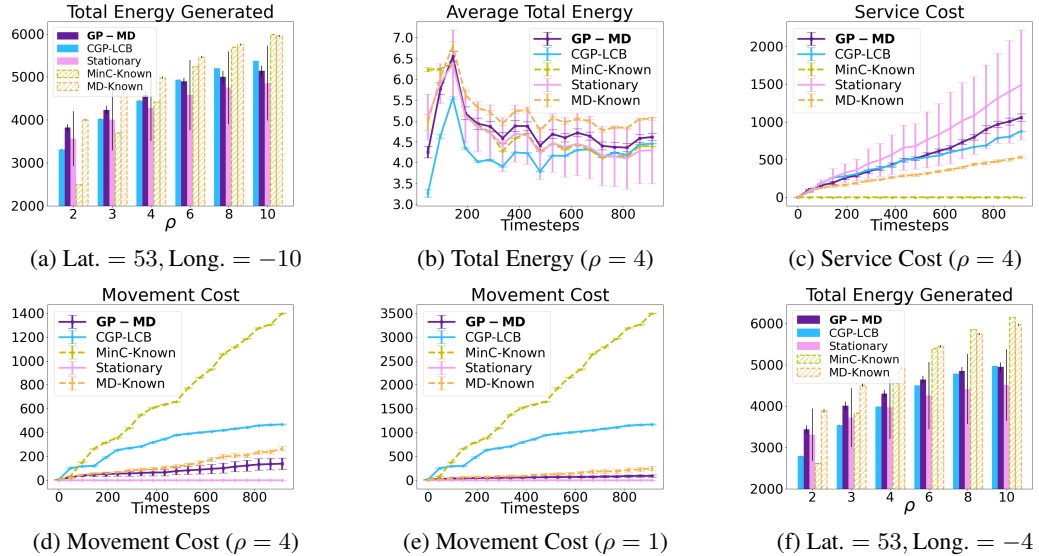

(a) Lat. $= 53$, Long. $= -10$

(b) Total Energy ($\rho = 4$)

(c) Service Cost ($\rho = 4$)

(d) Movement Cost ($\rho = 4$)

(e) Movement Cost ($\rho = 1$)

(f) Lat. $= 53$, Long. $= -4$

Figure 3: AWE altitude optimization task; Fig. 3a: Total energy generated for 960 hours based on the wind data at a single location (Latitude$= 53$ and Longitude$= -10$). GP-MD outperforms previously used CGP-LCB (that optimizes for service costs only) for a range of $\rho$ values that favor the service against the movement cost. Fig. 3b: The average total generated energy over 960 hours. Figs. 3c and 3d: The service and movement costs for $\rho = 4$. Fig. 3e: The movement costs for $\rho = 1$. The movement energy loss is slightly lower for GP-MD as compared to $\rho = 4$ due to higher importance towards movement cost reduction. Fig. 3f: Same as Fig. 3a, albeit by using wind data from a different location (Latitude$= 53$, Longitude$= -4$).

the energy lost in changing altitude (from $x$ to $x'$):

$$E_M(x, x') = c_3 V_r^2 |x - x'|, \tag{17}$$

where $c_3 = 0.15$ (see Appendix F for more details).[2]

We assume that a wind speed gauge is attached to the turbine, and the operator knows the wind speed at the current altitude. Hence, instead of directly learning $f(x, t)$, we learn $V_w(x, t)$ and use its confidence bounds to calculate the confidence bounds of $f(x, t)$. To learn $V_w(x, t)$, we normalize the inputs, and fit a GP with RBF kernel (lengthscale=3.67, outputscale=6.85 and noise parameter=2.73).

We run the algorithms for different $\rho$ for 960 timesteps, where again $\rho$ is used to multiply $E_S$. For each $\rho$, the algorithms were initiated with every possible starting point (25 different altitudes), and ran for 3 iterations. Based on this we plot the total energy generated w.r.t. varying $\rho$ in Fig. 3. In Figs. 3a and 3f, we show the performance of the algorithms at two different locations (we also consider additional locations and time periods in Appendix F). We use different values of $\rho > 1$ to show the robustness of our algorithm (as $\rho$ increases, the importance of the service cost w.r.t. the movement cost in the overall objective increases). Our algorithm outperforms CGP-LCB for a range of $\rho$ values. As $\rho$ keeps increasing, we observe that MINC-KNOWN closes the performance gap to MD-KNOWN, and the same is happening with CGP-LCB w.r.t. GP-MD. In Fig. 3b, we focus on a particular $\rho = 4$, and notice that GP-MD performs better than CGP-LCB and STATIONARY algorithm at this location. In Fig. 3c, we plot the service cost and observe that both learning algorithms GP-MD and CGP-LCB have lower service cost than the STATIONARY baseline. We also note that due to the implicit service cost definition, the MINC-KNOWN baseline achieves zero service cost. In Figs. 3d and 3e, we compare the movement energy loss for $\rho = 4$ and $\rho = 1$. As expected, $\rho = 1$ results in slightly lower GP-MD movement energy loss due to the tradeoff shifting towards the movement cost.

## 5 Conclusions

We have considered the problem of optimizing an unknown cost function subject to time-varying contextual information, as well as *movement costs* of changing the selected action from round to

---

[2]According to the power equation from [7], $E_M(x, x')$ would depend on $V_w(x, t)$, whereas, we assume our movement cost is based on a fixed metric and is independent of contexts. Hence, we simply approximate $V_w(x, t)$ by $V_r$ and consider time-independent movement costs.

round. Our problem formulation is motivated by Airborne Wind Energy systems, where one seeks to optimize the operating altitude of the wind turbine to maximize the amount of generated energy. We propose a novel algorithm, GP-MD, which makes use of GP confidence bounds and employs the mirror descent techniques from [19] for solving MTS problems. We analyze the theoretical performance of our algorithm by providing a rigorous regret bound. Moreover, we demonstrate its performance in synthetic experiments and on an AWE application by using real-world data. GP-MD carefully trades off service and movement costs while at the same time learning about the unknown objective function and yielding improved performance (i.e., generating more energy) compared to the considered baselines. Our setup and analysis open up multiple interesting directions for further exploration. For instance, an extension to continuous action spaces via discretization arguments is an immediate direction for future work. Another interesting direction is to analyze the single-episode setting and obtain general sublinear regret guarantees.

# 6 Acknowledgements

The authors would like to thank James R. Lee and Christian Coester for the various discussions regarding their paper [19] during the course of this work. This project has received funding from the European Research Council (ERC) under the European Unions Horizon 2020 research and innovation programme grant agreement No 815943.

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
