# OpenReview forum: "Movement Penalized Bayesian Optimization with Application to Wind Energy Systems"
_NeurIPS.cc/2022/Conference — NeurIPS 2022 Accept_

### Official Review · Reviewer_rb6B · 2022-07-10

**Rating:** 6
**Confidence:** 3
**Soundness:** 3 good
**Presentation:** 3 good
**Contribution:** 3 good

**Summary:**

This paper studies the problem of contextual Bayesian optimization that considers the cost of switching decisions at each round. The main idea is incorporating the online learning for metrical task systems and introducing a randomized mirror descent algorithm. The paper provides regret guarantees and evaluate on altitude optimization for Airborne Wind Energy Systems. Empirically, the proposed approach consistently outperforms baseline CBO approaches.

**Questions:**

- Why does the paper only considers CBO with moving cost, not the vanilla BO with moving cost? If vanilla BO with moving cost, would the problem be reduced to  multi-objective BO?
- Why does the experiment not compare to the method in paper "Bayesian optimization for modular black-box systems with switching costs"?

I would like to hear more justifications on these questions from the authors.

**Limitations:**

- No mention on open-sourcing the code.

**Strengths And Weaknesses:**

Strengths:

- Study an important and novel problem in contextual Bayesian optimization that considers moving cost between configurations.
- Good real-word case study.
- Writing is easy to follow.

Weaknesses:

- Possible missing comparison to other baseline.
- No code released.

Originality:

The originality of the paper is two folds. First, it proposes a new problem that has not been studied explicitly in the prior work. Second, it introduces a new method to solve this problem, although the majority of the proposed algorithm is derived from existing work by Coester and Lee.

Quality:

The problem is well defined and the proposed method is complete. The algorithm is not complicated which makes the approach accessible to potentially wide audience.

Clarity:

The writing is clear and easy to follow.

Significance:

The paper studies a very important problem in contextual Bayesian optimization that considers moving cost between configurations.

---

> ### Author Response · Authors · 2022-07-30
> **Response to Reviewer rb6B**
>
> We thank the reviewer for the overall positive assessment of our paper. Next, we provide our answers to the reviewer’s questions:
>
> – (baseline) The mentioned paper does not consider the contextual setting that is essential for our considered wind energy application and hence the two are not directly comparable. Moreover, it assumes that the system of interest has a modular structure (as detailed in their Section 3). In such a modular setting, the cost at each round is the number of modules that has to be changed rather than the amount of change of each variable. We will include the discussion on the differences with respect to this work in the related work section.
>
> – (vanila BO) Our setup is inspired by the real-world application of wind-energy systems that naturally fits the contextual BO setup due to the varying wind speeds. This setting is more difficult since the learner does not observe contextual information in advance. This is something that is crucially not present in the vanilla BO setting. Multi-objective BO typically assumes that all the objectives are unknown, while in our case the distance function is known.
>
> – (code) We will release the code and open-source it upon acceptance. Currently, the code can be found in the supplementary material.

---

> > ### Comment · Reviewer_rb6B · 2022-08-08
> > **Clarification on vanilla BO**
> >
> > "This setting is more difficult since the learner does not observe contextual information in advance. "
> >
> > I am a bit confused. Are you referring that CBO is more difficult compared to what? In CBO, how is it possible that the learner does not observe contextual information in advance? In abstract, it writes ". In this setting, the learner receives context (e.g., weather conditions) at each round"

---

> > > ### Author Response · Authors · 2022-08-08
> > > **Clarification on vanilla BO**
> > >
> > > We thank the reviewer for the response. We will try to further clarify this. Please let us know if it is still unclear.
> > >
> > > We acknowledge that in Contextual BO, the learner gets to observe the context at the current round before making the decision (this is what we also consider). However, in our previous response, we meant to highlight that the learner does not get to observe the entire sequence of contexts in advance.
> > >
> > > While this does not play a role in standard CBO, it is a major challenge in our setting due to the presence of movement costs. For example, assume at time $t$ the learner selects the optimal greedy action $a_t$ (in terms of service cost) given the observed context. Then, after observing the next context at time $t+1$, this may turn out to be a suboptimal choice since $a_t$ may be far away (as measured by the movement cost) from the optimal action for the context at $t+1$. Hence, deciding upon the optimal action sequence despite not knowing future contexts and in the presence of movement costs is a more difficult problem.

---

### Official Review · Reviewer_htxZ · 2022-07-10

**Rating:** 6
**Confidence:** 3
**Soundness:** 3 good
**Presentation:** 3 good
**Contribution:** 3 good

**Summary:**

The paper proposes and resolves the episodic contextual Bayesian optimization (BO) with movement costs by combining BO and an existing online learning solution to the metrical task system. The proposed algorithm, GP-MD, is proved to have a sublinear regret guarantee. Furthermore, GP-MD is empirically shown to outperform existing contextual BO in problems with movement costs, especially the real-world optimization problem for wind energy systems.

**Questions:**

Please clarify on the episodic setting in the experiments and scalability of the proposed approach mentioned above.

**Ethics Review Area:**

["I don’t know"]

**Limitations:**

The authors could improve the discussion on the scability of the work with respect to the size of the action and context spaces.

**Strengths And Weaknesses:**

The proposed problem of contextual BO with movement costs is new and significant in practice (e.g., with the applications to wind energy systems). Even though I have not thoroughly gone through the appendix, the proposed algorithm is sound and convincing. The empirical performance of GP-MD is shown in both synthetic and real-world optimization problems.

On the other hand, if I am not mistaken, the setting in the experiments is not episodic: in the algorithm and analysis, the GP posterior is updated after each episode, while in the experiments, the GP posterior is updated after every timestep and the experiments are performed in only 1 episode. To be consistent, the experiments should be performed in multiple episodes, or the algorithm and analysis should be done with GP posterior updated after every timestep. I am also curious about the real-world example where the episodic setting applies.

How is the proposed algorithm scalable to larger action and context space? When the dimension of the action and the context space is larger (e.g., 4 or 5), discretizing the space will result in a large number of points. Therefore, a discussion or the time complexity analysis of the proposed algorithm with respect to the size of the action and context spaces can be helpful.

In the real world experiment, the GP hyperparameters are assumed to be known (fixed), which is not very practical.

There is a minor typo: Ne -> Nep in line 132.

---

> ### Author Response · Authors · 2022-07-30
> **Response to Reviewer htxZ**
>
> We thank the reviewer for the overall positive assessment of our paper. Next, we provide answers to the specific questions:
>
> – (large action and context spaces) Our approach only considers finite action sets (as stated in the problem statement). The dimensionality of the context vectors also does not play a major role since we make use of kernels. Overall, choosing an action (updating the probabilities and sampling an action) at each round scales with $\mathcal{O}(n)$ (See also “computational complexity” response to reviewer UfT3). In general, the scalability will mimic the scalability of Bayesian optimization/ Gaussian Process.
>
> – (on episodic setting) We kindly refer the reviewer to the Response to Reviewer LNYH “on episodic setting” where we explain our decisions (also “on H dependence” response).

---

### Official Review · Reviewer_LNYH · 2022-07-11

**Rating:** 5
**Confidence:** 4
**Soundness:** 2 fair
**Presentation:** 3 good
**Contribution:** 3 good

**Summary:**

This paper proposes a contextual Bayesian optimization algorithm with penalisation for movement cost, which is motivated by the problem of tuning the altitude of wind turbines to maximize energy output while minimizing the altitude adjustment. That is, a movement cost is incurred which is larger if the difference between the actions selected in the current and the previous steps is larger. The proposed algorithm is based on the problem of metrical task systems, and combines lower confidence bound from Bayesian optimization with online mirror descent. The regret of the proposed algorithm is analysed, and the algorithm achieves competitive empirical performances in a real-world wind energy systems experiment.

**Questions:**

- Some of the questions have been listed under Weaknesses in "Strengths And Weaknesses" above.
- Equation (3), and lines 128-129: I'm curious about the connection between the regret definition here and the usually adopted regret definition in contextual BO/contextual bandit (e.g., reference [30] in the paper). Does the "offline optimal action sequence" also have the knowledge of the groundtruth functions of f and d? If we let $\alpha=1$ and $\beta=0$, will equation (3) reduce to the same regret definition as that in contextual BO/contextual bandit?

**Limitations:**

I didn't find discussions of limitation. I can imagine that potential limitations could be the dependence on $H$ (which I discussed at length under Strengths and Weaknesses), and that the action space is required to be discrete. The paper has no negative societal impact that I can think of.

**Strengths And Weaknesses:**

Strengths:
- The proposed problem of contextual BO with movement cost is well motivated, through the AWE systems. The experiment on AWE using real-world wind data is also a nice demonstration of the practicality of the proposed method.
- The idea of using the confidence bounds often adopted in BO as a bridge to connect BO with online learning algorithms is a very interesting and promising idea.
- The experiments are well designed, and the experimental results indeed show the advantage of the proposed method.
- The paper is well written.

Weaknesses:
- Two recent works on BO with movement cost are missing:
[1] SnAKe: Bayesian Optimization with Pathwise Exploration, 2022.
[2] Scaling Gaussian Process Optimization by Evaluating a Few Unique Candidates Multiple Times, 2022.
The methods in both of these papers are also motivated by the issue of movement cost (i.e., a larger cost is incurred if the selected inputs in two consecutive iterations have larger differences). Both papers were available in Jan 2022, and hence I believe they should have been discussed and compared with.
- I have a major concern regarding the dependence of the regret bound (Theorem 1) on the episode horizon H. Normally, in BO and other bandit algorithms, we consider the regret in every step. That is, if there are in total $T$ iterations/steps, the final theoretical result is usually presented as an upper bound on the total cumulative regret **across all $T$ steps**. In the context of this paper, it will translate to the total cumulative regrets across all $H\times N_{ep}$ steps. However, the regret bound in Theorem 1 has an undesirable dependence on H: for the squared-exponential kernel, both terms in the regret upper bound depend on $H$ linearly, which implies that the total regret in all $H\times N_{ep}$ steps is linear in the number of steps ; for the Matern kernel, the dependence of the first term on H will be super-linear, implying super-linear total cumulative regret.
- Related to the last point, lines 237-238 says that "a similar dependence on H is also present in the batch Bayesian optimization results and analysis that we also build upon (see [19, Theorem 2])". I understand what is meant here, but I think this statement is misleading. In [19], their regret bound has a dependence on $\exp(\gamma_H)$ ($H$ will correspond to the batch size in [19]) rather than $H \times \exp(\gamma_H)$. Moreover, the regret bound depending on $\exp(\gamma_H)$ is only their intermediate results, and in their subsequent analysis, they had shown that by following a special initialization procedure, the dependence on $\exp(\gamma_H)$ can be replaced by a constant which is independent of $H$. Therefore, I think [19] should not be used to justify the dependence of the regret bound on $H$ in this paper. Please correct me if I had any misunderstanding here.
- Also related to the point above regarding the dependence on $H$: lines 263-264 say that "We run the algorithms over a single episode" and that "We also discover that updating the confidence bounds after every timestep in GP-MD leads to better performance in practice". Therefore, it seems that in your practical implementation, you have entirely ignored the episodic setting and reverted back to the standard contextual BO setting. If I understand correctly, (if you ignore the heuristic of updating the confidence bounds after every timestep) in this setting, $H$ will be equal to $T$, and hence your regret bound will not be sub-linear anymore as I discussed above in the point on the dependence on $H$.
- Given what I have discussed in the three points above, I think an explanation as to "why the theoretical analysis follows the episodic setting" is needed. Because for now, it seems to me that the episodic setting is only adopted so that the final regret upper bound can be sub-linear in the total number of episodes, while in practice, the episodic setting is not needed.
- (minor) The paper is in general well written, including the method section of Section 3. The only place I think needs more explanations/intuitions is equation (12) and the paragraph following it.

---

> ### Author Response · Authors · 2022-07-30
> **Response to Reviewer LNYH**
>
> We thank the reviewer for the outlined strengths of our paper and the insightful review. Next, we answer every question posed by the reviewer:
>
> – (on references) We thank the reviewer for the provided relevant references. Both works are recently presented at the ICML2022 conference. We will include the discussion regarding the differences to these works. In particular, the first work lacks theoretical guarantees, while the second work does not include the cost in the regret definition. Both works do not consider the contextual setting that is essential in our case and the application that we consider.
>
> – (on $\exp(\gamma_H)$) We thank the reviewer for pointing this out. We have addressed this issue above by following a different analysis technique akin to [17] that eliminates the dependence on $\exp(\gamma_H)$. We have made the changes accordingly. Please see Eqs. 66-71 for the analysis steps and Theorem 1 for the updated result.
>
> – (on episodic setting) The reviewer is correct in that the episodic setting allows us to prove sublinear regret bounds in the number of episodes. Intuitively, this is because it yields more stable learning when coupled with the mirror descent subroutines. The episodic setting is also of practical interest since real-world systems (such as wind turbines) are often reset to a given initial condition at the end of each day, and it is more convenient to update the GP model with batches of data. Nevertheless, in our simulated experiments, we have performed continuous updates (for faster learning) and empirically observed sublinear regrets also in such a case.
>
> – (H dependence) The reviewer is correct that the dependence on H in our bound is linear. We note that our setting bears some similarities (e.g., initial state, state transitions) to the episodic kernelized RL problem that has a similar dependence on H (e.g., see Theorem 1 in [17]). On the other hand, in the non-episodic setting, our preliminary analysis shows $\mathcal{O}(\gamma_{T} T^{2/3})$ bound (with minor modifications to the algorithm and analysis), which is still sublinear for some kernels. However, we note that achieving the standard stochastic BO regret rates is not to be expected here since our setting (with movement costs and possibly adversarial contexts) is significantly more challenging. We are happy to include this discussion and proof in the supplementary material (or share it here in the subsequent post).
>
> – (on connection between regret definitions) The offline optimal action sequence is obtained assuming both the knowledge of $f$, $d$, and all the contexts in advance. Unlike the contextual BO works, we additionally consider the movement costs in Eq. 2. The latter changes the problem crucially because the current decision also influences future regret. This is not the case in the standard contextual BO (this is also why the initial action is important). Moreover, we also allow the contexts to be adversarially generated.
>
> – (Eq. 12 - minor) Given additional available space to incorporate reviewers’ feedback, we will provide more intuition on Eq. 12.
>
> We have answered all of the questions provided by the reviewer.  We firmly believe that none of these points amount to any significant changes being required in the paper, and we hope that the reviewer will reconsider updating the provided rating. Please feel free to let us know if you have any further questions or concerns.

---

> > ### Comment · Reviewer_LNYH · 2022-08-08
> > **Response to Authors**
> >
> > I'd like to thank the authors for the response.
> >
> > - The authors' discussion of the references makes sense, since there indeed seems to be important differences.
> > - (on $\exp(\gamma_H)$): I find this solution unsatisfactory, because according to the updated paper, although the dependence on $\exp(\gamma_H)$ has been removed, it seems to have been replaced by a dependence on $H$ (yet the reason why $\exp(\gamma_H)$ was a concern to me was because it causes a bad dependence on $H$). As a result, the resulting dependence on $H$ is still linear. For the squared exponential kernel, this doesn't really change the dependence on $H$.
> > - (on episodic setting): I understand that the episodic setting is practical, but this doesn't remove the discrepancy between the analyzed episodic setting and the experiments.
> > - (H dependence): I find the discussion here encouraging. An analysis of the regrets in the non-episodic setting would be a nice complement to the main analysis in my opinion, and I do encourage the authors to include it in the paper. And yes, I understand the regret growth rate in the episodic setting may be worse, but fortunately the growth rate of $\mathcal{O}(\gamma_T T^{2/3})$ is still sub-linear at least for the squared exponential kernel. A valuable aspect of adding this analysis is that the analyzed setting would be more consistent with the experiments.
> > - The other two points also make sense.
> >
> > All things considered, I'll increase my score to 5.

---

> > > ### Author Response · Authors · 2022-08-08
> > > **Response**
> > >
> > > We thank the reviewer for going through our rebuttal in detail, acknowledging all our responses, and adapting the score. As suggested by the reviewer, we will add the analysis for non-episodic/single episode setting with all the intricate details.

---

### Official Review · Reviewer_UfT3 · 2022-07-14

**Rating:** 7
**Confidence:** 2
**Soundness:** 4 excellent
**Presentation:** 4 excellent
**Contribution:** 3 good

**Summary:**

This paper provides an algorithm for the problem of episodic Contextual Bayesian optimization (CBO) under “switching costs” associated with the decision at each round. This problem formulation is motivated by the application of altitude optimization for Airborne Wind Energy (AWE) systems, in which a wind turbine’s operating altitude can be changed between episodes in order to maximize energy production, but where the changing of altitude itself uses energy. Noting that this problem setup is similar to that in metrical task systems (MTSes) except that the objective function is unknown, the authors propose a Gaussian Process-based variation of the mirror descent schemes used to address MTS problems. They provide a bound on the cumulative regret of their algorithm, and show in both synthetic experiments and experiments in the AWE setting that their algorithm outperforms algorithms that neglect switching costs.

**Questions:**

* In the AWE experiments, in the real-world setting, is $f$ known or unknown? If it is known, what is the justification for the current problem setup? If unknown, how are the MinC-Known and MD-Known baselines run for this experiment?
* What is the implication of assumptions such as the regularity assumptions on $f$, which are utilized to construct the proposed method? How does that affect in what settings (related to wind energy production or more broadly) the method may not be expected to work?
* How computationally intensive is the method, and how does that affect what settings it can be used in and/or how well it might scale?

**Limitations:**

* No limitations are discussed. However, at minimum, limitations associated with the assumptions made in the paper as well as (e.g.) computational complexity considerations associated with using GPs should be discussed.
* In the checklist, the authors choose “N/A” when asked to describe any potential negative societal impacts of their work. However, as the proposed GP-MD algorithm is relatively general, it is of course possible that it can have bad applications in addition to the good ones (AWE) focused on in the paper.


**Strengths And Weaknesses:**

Strengths:
* The paper is very well presented. Even as someone who is outside this area, I was easily able to understand the broad strokes of the motivation and approach (even if not the details).
* The problem formulation seems well-motivated by a problem of societal importance, and per the authors’ claim is not previously addressed in the literature (though I am not familiar enough with the literature to validate this claim).
* The proposed method is a nice synthesis of ideas proposed separately in the CBO and MTS setting.
* The experiments are thorough and convincing, demonstrating and validating the settings in which the proposed algorithm and baselines do/don’t do well.

Weaknesses:
* In Section 4.1 (AWE experiments), it is not clear to me how MinC-Known and MD-Known were run as baselines if $f$ is in reality not known. Alternatively, if $f$ is in reality known in the AWE setting, then it is not clear to me why the proposed method is needed. More clarity on this would be helpful in understanding whether the current approach is indeed needed and/or how the experiments were run.
* The authors do not meaningfully discuss any limitations of their approach. However, at minimum, limitations associated with the assumptions made in the paper as well as (e.g.) computational complexity considerations associated with using GPs should be discussed.

Minor points:
* Would it be possible to add error bars to Figure 1?

---

> ### Author Response · Authors · 2022-07-30
> **Response to Reviewer UfT3**
>
> We thank the reviewer for the positive assessment of our paper. We summarize the main answers as follows:
>
> –  (on  $f(\cdot)$ ) In practice, $f(\cdot)$ is unknown and can only be estimated from previously observed data (as proposed in our approach). For benchmarking purposes, we have simulated $f(\cdot)$ using the hourly wind data *estimated a-posteriori*. Hence, MinC-Known and MD-Known are ideal but unrealistic baselines that we run to demonstrate the upper-bound on the performance achievable by the learning approaches.
>
> –  (on regularity assumption) The regularity assumptions implies that $f$ is smooth (i.e., no drastic variation in function values for small changes in the variables). For example, in our case, the service cost should vary smoothly with respect to changes in wind strength/turbine altitude.
>
> – (computational complexity) The HST tree construction with $n$ actions is $\mathcal{O}(n^{2})$. The updates at each round (Lines 7-9 in GP-MD) iterate through every vertex in the tree and hence will be $\mathcal{O}(n)$. Due to the HST structure, the sampling from conditional optimal coupling distribution in Line 13 is $\mathcal{O}(n)$. Overall, choosing an action (updating the probabilities and sampling an action) at each round scales with $\mathcal{O}(n)$.
>
> – (limitations) Besides the well-known limitations of GPs, our work is concerned with the finite discrete action space. Extensions to the continuous action spaces are possible via discretization arguments and are left for future work. We will further highlight this and update the checklist.
>
> – (minor point) Figure-1 was just to showcase the preliminary ideas of our approach. The technically correct versions of the plots (with error bars) are detailed in Section 5. Nevertheless, we will update the plot with error bars.

---

> > ### Comment · Reviewer_UfT3 · 2022-08-07
> > **Response**
> >
> > Thank you to the authors for their response, in particular for the clarification on whether $f(\cdot)$ is known.

---

### Author Response · Authors · 2022-08-08
**Author-Reviewer Discussion**

Dear reviewers,

The author-reviewer discussion period is ending in less than two days. We kindly request your acknowledgment of our responses (if you haven't done so already), and that you let us know if there are any issues that you still find problematic, and/or check that your score is in agreement with your updated understanding of our work.

Thank you for your time and consideration.

---

### Meta-Review · Area_Chair_93MX · 2022-08-24

**Recommendation:** Accept
**Confidence:** Certain

**Metareview:**

This paper proposes a contextual Bayesian optimization algorithm with penalization for movement cost, which is motivated by the problem of tuning the altitude of wind turbines to maximize energy output while minimizing the altitude adjustment. That is, a movement cost is incurred which is larger if the difference between the actions selected in the current and the previous steps is larger. The proposed algorithm is based on the problem of metrical task systems, and combines lower confidence bound from Bayesian optimization with online mirror descent. The regret of the proposed algorithm is analyzed, and the algorithm achieves competitive empirical performances in a real-world wind energy systems experiment.

All reviewers agree that this is an important and novel problem for Contextual Bayesian Optimization.  The experiment were a nice demonstration of the practicality of the approach.  All four reviewers were on the positive side for acceptance.


**Award:**

No

---

### Decision · Program_Chairs · 2022-09-14

Accept